# Cardiac findings in infants with in utero exposure to Zika virus – a follow up longitudinal study

Brian N. Dang[1,2], Karen Kikuta[3], Trevon Fuller[4], Patricia Brasil[4], Zilton Vasconcelos[5], Dulce H. G. Orofino[6,7], Maria Elizabeth L. Moreira[6], Karin Nielsen-Saines[8]*

1 Department of Pediatrics, University of California, San Francisco, California, United States of America, 2 Department of Pediatrics, Division of Cardiology, Lucille Packard Children's Hospital at Stanford University, Palo Alto, California, United States of America, 3 David Geffen School of Medicine at the University of California, Los Angeles, California, United States of America, 4 Department of Acute Febrile Illnesses, Evandro Chagas National Institute of Infectology, Oswaldo Cruz Foundation, Rio de Janeiro, Brazil, 5 Department of Clinical Immunology, Fernandes Figueira Institute, Oswaldo Cruz Foundation, Rio de Janeiro, Brazil, 6 Department of Pediatrics, Fernandes Figueira Institute, Oswaldo Cruz Foundation, Rio de Janeiro, Brazil, 7 Souza Marques School of Medicine, Rio de Janeiro, Brazil, 8 Department of Pediatrics, Division of Infectious Diseases, David Geffen School of Medicine at the University of California, Los Angeles, California, United States of America

* knielsen@mednet.ucla.edu

## Abstract

### Background

Zika virus (ZIKV) is primarily known for its impact on the fetal central nervous system potentially leading to Congenital Zika Syndrome (CZS). Emerging evidence suggests ZIKV may also affect cardiac development. We conducted a follow-up study evaluating cardiologic findings in infants from ZIKV-exposed mothers.

### Methods

Infants born to mothers with PCR-confirmed ZIKV infection during pregnancy and/or who had positive ZIKV PCR results at birth received echocardiograms in the first year of life. Repeat imaging within 12 months was requested for infants with identified abnormalities. Frequencies of cardiovascular (CV) abnormalities were evaluated using Pearson $\chi^2$ test, Fisher's exact test, and descriptive statistics. Predictors of CV abnormalities were assessed using multivariate logistic regression, as well as univariate and multivariate prevalence estimates. Sensitivity analysis assessed the robustness of associations when stratified by age at echocardiography (early vs late).

### Results

One hundred sixty-nine children with antenatal ZIKV-exposure had echocardiograms; 30.8% were microcephalic (MC). Thirty (17.8%) had cardiac anomalies. MC children had a higher frequency of CV abnormalities than normocephalic (NC) children (26.9% vs 13.7%, p = 0.04). Twenty-four of 30 children (80.0%) returned for repeat

**Data availability statement:** All relevant data are in the manuscript and its supporting information files.

**Funding:** This work was supported by the National Institutes of Health (grant numbers AI140718, AI129534, AI172252 and EY028318 to K.N.S.); Brazilian National Council for Scientific and Technological Development, CNPq (grant numbers 311657/2023-5, 441098/2016-9 and 305090/2016-0 to M.E.L.M.) and Carlos Chagas Filho Foundation Research Support for the State of Rio de Janeiro, FAPERJ (grant number E_18/2015TXB. to M.E.L.M.); Doximity Foundation Grant to B.N.D., UCSF Resident Clinical and Translational Research Funding Award to B.N.D. The funders had no role in study design, data collection and analysis, decision to publish, or preparation of the manuscript.

**Competing interests:** The authors have declared that no competing interests exist.

imaging; of that group, 25.0% continued to demonstrate defects. Rates of persistent defects between the MC vs. NC cohorts were 33.3% vs 16.7%, respectively (p = 0.64). Presence of CV defects was significantly associated with MC (OR=3.40, 95% CI 1.15-10.02; p = 0.03). Among those with echocardiography performed later, MC was still associated with higher risk of abnormalities (OR=6.0, 95% CI 1.03-34.94; p = 0.046).

## Conclusions

A higher frequency of cardiac defects was noted in ZIKV-exposed infants than the general population. Most defects resolved on follow-up. The presence of a congenital heart defect (CHD) could be considered a parameter of CZS given its association with MC.

## Author summary

Zika virus (ZIKV) is best known for affecting the fetal brain and causing Congenital Zika Syndrome (CZS), but growing evidence suggests it may also impact heart development. We studied infants born to mothers with PCR-confirmed ZIKV infection during pregnancy or infants who tested positive at birth, and all received echocardiograms in their first year. Those with abnormalities were asked to return for repeat imaging. We analyzed how often heart findings occurred and whether certain factors, like microcephaly (MC), were linked to these abnormalities. Among 169 ZIKV-exposed infants, 30% had MC and 17.8% had heart defects. Children with MC had a higher rate of abnormalities than those without MC (26.9% vs 13.7%). Of the infants who returned for repeat imaging, most defects resolved, although persistent defects were more common in the MC group. MC was significantly associated with a higher chance of having a heart defect, even when looking at infants who had later echocardiograms. Overall, ZIKV-exposed infants had a higher rate of heart defects than expected in the general population, and while most findings improved over time, the presence of a congenital heart defect may be considered another feature of CZS.

## Introduction

Zika virus (ZIKV) is a mosquito-borne flavivirus first identified in 1947 in a febrile Rhesus macaque in Uganda [1,2]. In 2007, ZIKV was found to be a significant human pathogen, as outbreaks were recorded worldwide in Africa, the Americas, the Pacific islands, and Asia [1,3]. In 2015, cases of a "dengue-like syndrome" began to emerge in northeastern Brazil, later identified as ZIKV by reverse transcription polymerase chain reaction (RT-PCR) and confirmed by DNA sequencing [3]. The epidemic quickly spread throughout the country and by the end of 2016, more than 200,000 cases were reported [4].

ZIKV exposure and infection was found to lead to a variety of long-term consequences that are still being explored. Clinical manifestations are often mild in adults, but the virus can have severe teratogenic repercussions, impacting fetal, neonatal and pediatric neurodevelopment [3,5–13]. During the Brazilian epidemic, ZIKV was identified in pregnant patients who had infants with microcephaly (MC) via RT-PCR in blood and urine samples, amniotic fluid, and tissue samples from fetuses of women who presented with rash during pregnancy [8,14]. A broad spectrum of adverse neurologic outcomes has since been described among infants with antenatal ZIKV exposure, including seizures, developmental delay, auditory and visual impairments, abnormal tone, feeding difficulties, and characteristic brain imaging abnormalities [6–13,15–20]. These findings constitute Congenital Zika Syndrome (CZS), a condition marked by severe neurological, structural, and ophthalmologic abnormalities [7,9–13,15–23]. Of note, children with antenatal exposure to ZIKV may not necessarily have MC and CZS, but can still have a more subtle manifestation of disease.

In addition to neurosensory repercussions, a cross sectional study by Orofino et al. (2018) demonstrated that laboratory-confirmed antenatal exposure to ZIKV was associated with cardiac defects identified by transthoracic echocardiography [24]. While the lesions were not severe, ZIKV-exposed infants were found to have a 10.8% prevalence of major structural heart defects. This rate was significantly higher than the ~1% frequency reported in the general infant population [25–29], although some suggest the rate may be as high as 7.5% with inclusion of all minor anomalies [30]. Here we present a follow-up cardiologic study of children with antenatal ZIKV exposure followed at a large pediatric referral center in Rio de Janeiro, Brazil.

In this longitudinal assessment, our research questions were: what is the prevalence and spectrum of structural cardiac abnormalities; do these risks differ by cephalic status at birth; and what proportion of anomalies persist on repeat echocardiography? Accordingly, our objectives were to (i) estimate the prevalence of any structural cardiac abnormalities in early infancy; (ii) compare prevalence and abnormality patterns between MC and normocephalic (NC) infants; (iii) characterize lesion-specific frequencies; (iv) determine the proportion of defects that resolve versus persist on follow-up; and (v) explore clinical and perinatal predictors of cardiac abnormalities.

## Methods

### Ethics statement

All parents or guardians provided written informed consent. Data of all study subjects followed appropriate de-identification to remove protected health information (PHI) per HIPAA Laws. The study was approved by the Institutional Review Board (IRB) of the Brazilian National Institute of Infectious Diseases (INI and IFF/FIOCRUZ #2675616.0.0000.5269) and UCLA (# 17–000104).

### Study population

We performed a longitudinal study of infants born during the Brazil ZIKV epidemic between 2015–2016. Children were followed at the outpatient Pediatric Infectious Disease Clinic at the Fernandes Figueira Institute (IFF-FIOCRUZ), the major referral center in Rio de Janeiro for suspected ZIKV cases during pregnancy. Although prenatal care and deliveries frequently occurred at outside facilities, families were referred to IFF-FIOCRUZ for postnatal follow-up related to ZIKV exposure and comprehensive evaluation, including echocardiography.

Study subjects were initially selected based on data from maternal and infant medical records, including maternal history of rash during pregnancy and results of maternal and infant specimens. Exposure to ZIKV was confirmed via testing in the mother and/or infant with RT-PCR of maternal blood or urine samples and/or amniotic fluid, as well as in fetal urine, serum, and/or cerebrospinal fluid (CSF) samples, and ZIKV IgM serologies in infant blood and CSF (21). Clinical data including gestational age at birth (pre-term <37 weeks), sex, birthweight (small for gestational age defined as < 10th percentile at birth), trimester of maternal infection during pregnancy, maternal age, delivery method, maternal hypertension,

and maternal diabetes were abstracted from the medical charts. Echocardiograms were offered to all children who had laboratory confirmation of maternal ZIKV infection in pregnancy. Participants who did not return for echocardiograms were excluded. The primary reasons for loss to follow-up were attributed to socioeconomic barriers commonly faced by low-income and socially vulnerable populations. Study subjects were screened for other congenital infections including HIV, cytomegalovirus, hepatitis B, hepatitis C, toxoplasmosis and rubella virus, as well as genetic syndromes associated with congenital heart defects (CHDs). Overall, no children were excluded since patients were referred to IFF-FIOCRUZ after meeting all inclusion criteria: primary ZIKV exposure without an underlying genetic syndrome or other congenital infection, and need for cardiovascular assessment. Blinding of cardiologists was not possible, particularly in children with microcephaly, where clinical features were evident. Further, echocardiographic evaluations were conducted during a time of increased clinical awareness of ZIKV-related findings. Infants were further characterized as either MC or NC, based on head circumference at birth. We considered microcephaly as head circumference at least two standard deviations below the mean for gestational age.

### Transthoracic echocardiography

Cardiologic assessment was conducted by pediatric cardiologists at IFF-FIOCRUZ. Imaging was performed without sedation, with full 2D and M-mode echocardiography with pulsed and continuous Doppler, as well as color Doppler, using the Siemens Acuson X300 Echocardiography system.

Initial echocardiograms were performed within the first year of life (Fig 1). Defects were stratified as minor or major according to the classification proposed by Orofino et al. 2018 [24] based on the hemodynamic significance of the lesion. Minor defects were defined as mild pulmonary branch stenosis (PBS) in term infants, mild tricuspid regurgitation (TR), and patent ductus arteriosus (PDA) in term infants up to 15 days of age or pre-term infants up to three months of age. Of note, in contrast to our prior paper, we did not consider persistent foramen ovale (PFO) to be a minor defect in the present study, since this can be an incidental finding in otherwise healthy children and is estimated to be present in approximately 25% of the population [31]. PBS in preterm infants was considered a normal finding. The presence of multiple minor defects, and all other echocardiogram defects, including ventricular septal defect (VSD), atrial septal defect (ASD), pulmonary hypertension (PH), persistent PDA, left ventricular hypertrophy (LVH), coarctation of the aorta (CoA), and bicuspid aortic valve (BAV) were considered major structural defects. Congenital defects requiring medical or surgical intervention within the first days of life were defined as severe. Study subjects were followed longitudinally after initial selection. Those with cardiovascular abnormalities identified on initial echocardiogram underwent repeat evaluation within 1–12 months. Infants with normal findings in infancy did not have repeat imaging performed.

### Statistical analyses

Patient demographics, clinical manifestations, and echocardiogram results were converted to binomial values and analyzed using Pearson $\chi^2$ test or Fisher's exact test (two-sided) when appropriate, with a p-value <0.05 considered significant. The results of initial and repeat imaging between the MC and NC cohorts were analyzed using Pearson $\chi^2$ test and Fisher's exact test, respectively. The absolute count and relative frequencies of each minor and major defect, stratified by cephalic status, were further delineated for both initial and repeat studies. A multivariate logistic regression analysis was performed to identify associations between cardiac defects in antenatally exposed infants and clinical/demographic predictors (gestation, sex, birthweight, delivery method, trimester of infection, mother's age at delivery, maternal hypertension, and maternal diabetes), as well as clinical outcomes (presence of microcephaly). Univariate and multivariate prevalence estimates were calculated. Sensitivity analysis assessed the robustness of associations when stratified by age at echocardiography. To this end, participants were classified as above or below the median age at echocardiography, denoted the "early" and "late" groups, respectively. All analyses were performed with SPSS, version 25.0 (SPSS Inc).

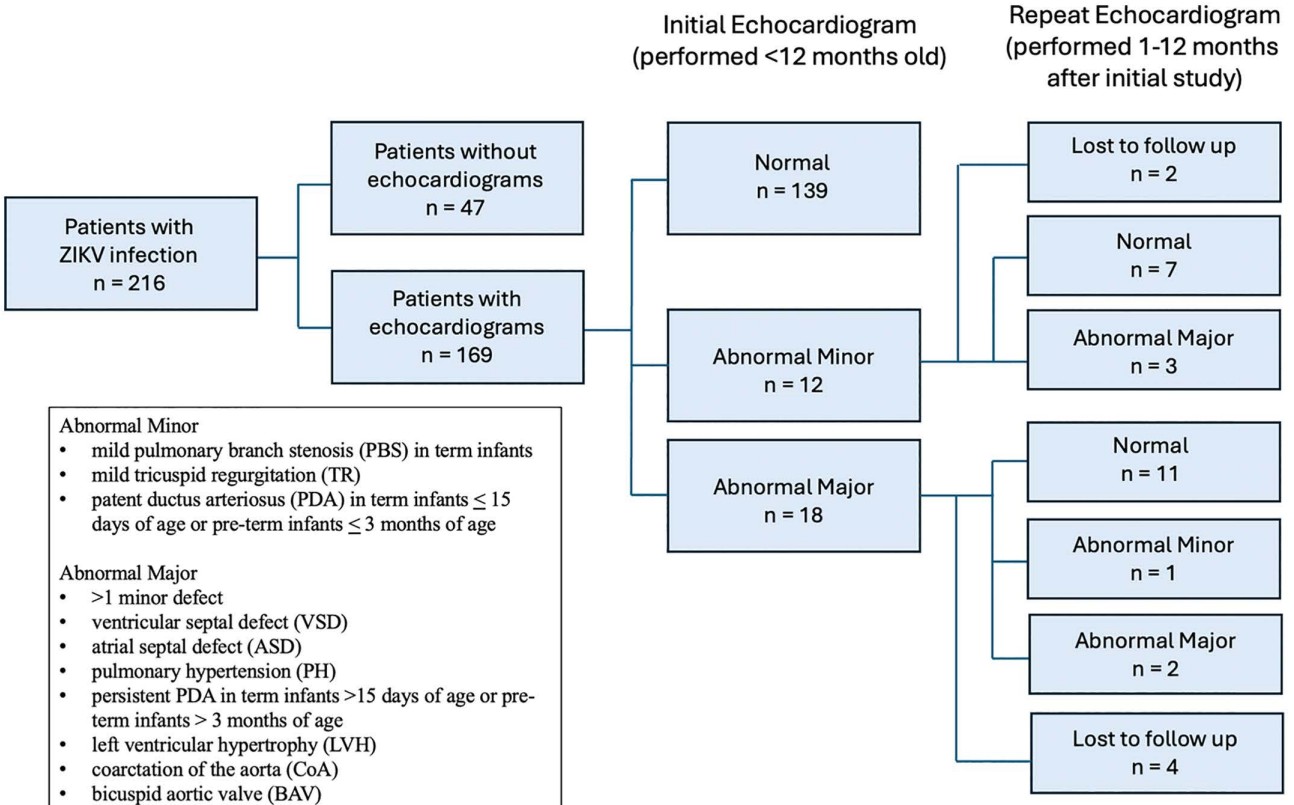

**Fig 1. Flow Diagram of Echocardiogram Assessments in Individual Patients.** Of 216 exposed infants, 169 had an initial echocardiogram < 12 months old: 139 normal, 12 abnormal minor, 18 abnormal major; 47 were lost to follow-up and not imaged. Of the 30 abnormal initial studies, 24 returned 1–12 months later: 18 normalized, 1 remained minor, 5 remained major; 6 were lost to follow-up.

## Results

There were 216 infants with laboratory confirmed ZIKV infection in pregnancy who had follow-up after birth at IFF-FIOCRUZ. In total, 169 children (78%) completed echocardiogram evaluation, and 47 children did not receive imaging as they were lost to follow-up (Table 1 and Fig 1). Most infants were normocephalic (69.2%), and the remaining 30.8% had microcephaly noted at birth. The distribution of males and females was roughly equal, and most infants (84.0%) were born at term. The majority of infants had a birth weight appropriate for gestational age (76.3%) and were delivered via Cesarean-section (67.5%). Most mothers were young (< 35 years of age) (79.9%), did not have hypertension (86.4%), and did not have diabetes (96.9%). MC children were significantly more likely to be born small for gestational age (SGA), to mothers under 35 years of age, and via vaginal delivery. The trimester of ZIKV and further breakdown of demographic variables are presented in Table 1.

In total, thirty infants (17.8%) had cardiac anomalies identified via echocardiogram. The mean age of children at the time of initial cardiac evaluation was 86 ± 80 days (Table 2). There were several infants with more than one cardiac defect identified, and the data presented in Table 2 represents the number of cardiac defects observed (e.g., one infant had a PDA and VSD, which contributed a count to both categories). Specifically, three MC and three NC children had two defects identified in each child. When comparing echocardiograms between children with MC and NC, a statistically significant higher incidence of abnormal cardiac findings was noted in MC children compared with NC children. The

**Table 1. Patient Demographics.**

| | Total | Normocephalic | Microcephalic | p-value |
|---|---|---|---|---|
| Number of patients (%) | 169 | 117 (69.2) | 52 (30.8) | |
| Gestation n (%) | | | | |
| Term | 142 (84.0) | 97 (82.9) | 45 (86.5) | 0.55 |
| Preterm | 27 (16.0) | 20 (17.1) | 7 (13.5) | |
| Sex n (%) | | | | |
| Male | 84 (49.7) | 58 (49.6) | 26 (50.0) | 0.96 |
| Female | 85 (50.3) | 59 (50.4) | 26 (50.0) | |
| Birth Weight n (%) | | | | |
| Appropriate for gestational age | 129 (76.3) | 104 (88.9) | 25 (48.1) | < 0.01 |
| Small for gestational age | 40 (23.7) | 13 (11.1) | 27 (51.9) | |
| Trimester Infection n (%) | | | | |
| Unknown/Asymptomatic | 12 (7.1) | 3 (2.6) | 9 (17.3) | <0.01 |
| 1st | 64 (37.9) | 29 (24.8) | 35 (67.3) | |
| 2nd | 69 (40.8) | 63 (53.8) | 6 (11.5) | |
| 3rd | 24 (14.2) | 22 (18.8) | 2 (3.9) | |
| Maternal Age n (%) | | | | |
| <35 years old | 135 (79.9) | 84 (71.8) | 51 (98.1) | < 0.01 |
| >35 years old | 34 (20.1) | 33 (28.2) | 1 (1.9) | |
| Delivery n (%) | | | | |
| Vaginal | 55 (32.5) | 29 (24.8) | 26 (50.0) | <0.01 |
| Cesarean-section | 114 (67.5) | 88 (75.2) | 26 (50.0) | |
| Maternal Hypertension n (%) | | | | |
| No | 140 (86.4) | 94 (83.9) | 46 (92.0) | 0.17 |
| Yes | 22 (13.6) | 18 (16.1) | 4 (8.0) | |
| Maternal Diabetes Mellitus n (%) | | | | |
| No | 157 (96.9) | 107 (95.5) | 50 (100.0) | 0.33 |
| Yes | 5 (3.1) | 5 (4.5) | 0 (0.0) | |

univariate prevalence estimates were 26.9% for MC versus 13.7% for NC (p = 0.04). The multivariate estimates adjusting for confounders were 31% for MC versus 12% for NC. When stratifying by major versus minor anomalies, 13.5% (7/52) of children with MC had major findings, which was similar to the rates of 9.4% (11/117) in the NC group (Fig 2). There were no severe defects (those requiring immediate medical or surgical intervention). The distribution of major defects was not significantly different between the two cohorts (p = 0.29), while the distribution of minor defects was borderline significant (p = 0.05). In terms of minor defects, the most common finding was a PDA (52.9%) for all infants and also stratified by MC or NC. Overall, there were four instances of mild TR and PBS each. Regarding major defects, the most common pathologies were VSDs (31.6%) and secundum ASDs (31.6%). There were two counts of LVH and persistent PDA each. We only found one instance of PH, CoA, and BAV. For the MC group, a VSD (37.5%) was the most common defect, and ASD (25%) was second. In the NC group, an ASD (36.4%) was the most common, followed by a VSD (27.3%). A further breakdown of cardiac findings between MC and NC patients is detailed in Table 2.

Infants with initially abnormal echocardiograms (n = 30) were referred for repeat imaging, with 24 (80%) returning for a second echocardiographic evaluation. Six children (25.0%) continued to demonstrate defects on reassessment. The remaining eighteen children (75.0%) no longer had identifiable defects on follow-up imaging, indicating resolution of cardiac abnormalities. The mean age of children who underwent repeat imaging was 188 ± 169 days (Table 2). The

**Table 2. Echocardiogram defects observed.**

| | Total | Normocephalic | Microcephalic | p-value |
|---|---|---|---|---|
| **Initial Echocardiogram number of patients (%)** | | | | |
| Normal | 139 (82.2) | 101 (86.3) | 38 (73.1) | 0.04 |
| Abnormal | 30 (17.8) | 16 (13.7) | 14 (26.9) | |
| Age in days, mean ± SD | 86 ± 80 | 98 ± 82 | 58 ± 68 | |
| **Minor defects noted (%)** | | | | |
| PDA | 9 (52.9) | 4 (50.0) | 5 (55.6) | |
| TR | 4 (23.5) | 2 (25.0) | 2 (22.2) | |
| PBS | 4 (23.5) | 2 (25.0) | 2 (22.2) | |
| Total defects | 17 | 8 | 9 | 0.05 |
| **Major defects noted (%)** | | | | |
| VSD | 6 (31.6) | 3 (27.3) | 3 (37.5) | |
| ASD | 6 (31.6) | 4 (36.4) | 2 (25.0) | |
| PH | 1 (5.3) | 0 | 1 (12.5) | |
| Persistent PDA | 2 (10.5) | 1 (9.1) | 1 (12.5) | |
| LVH | 2 (10.5) | 1 (9.1) | 1 (12.5) | |
| CoA | 1 (5.3) | 1 (9.1) | 0 | |
| BAV | 1 (5.3) | 1 (9.1) | 0 | |
| Total defects | 19 | 11 | 8 | 0.29 |
| **Repeat Echocardiogram number of patients (%)** | | | | |
| Normal | 18 (75.0) | 10 (83.3) | 8 (66.7) | 0.64 |
| Abnormal | 6 (25.0) | 2 (16.7) | 4 (33.3) | |
| Age in days, mean ± SD | 188 ± 169 | 180 ± 174 | 196 ± 168 | |
| **Minor defects noted (%)** | | | | |
| TR | 1 (100.0) | 1 (100.0) | 0 | |
| **Major defects noted (%)** | | | | |
| VSD | 2 (33.3) | 1 (100.0) | 1 (20.0) | |
| ASD | 1 (16.7) | 0 | 1 (20.0) | |
| Persistent PDA | 3 (50.0) | 0 | 3 (60.0) | |
| Total defects | 6 | 1 | 5 | 0.01 |

Abbreviations: PDA, patent ductus arteriosus; TR, tricuspid regurgitation; PBS, pulmonary branch stenosis; VSD, ventricular septal defect; ASD, atrial septal defect; PH, pulmonary hypertension; LVH, left ventricular hypertrophy; CoA, coarctation of aorta; BAV, bicuspid aortic valve

mean age stratified among the MC (196 ± 168 days) and NC (180 ± 174 days) infants was compared using a t-test and found not to be significantly different (p = 0.88). The six infants (20.0%) that did not return for evaluation were lost to follow-up. Two MC children with abnormal initial echocardiograms did not return for follow up, and their lesions included an ASD, PH, and PBS. Four NC children had no repeat imaging, with initial lesions being TR, PBS, ASD, and BAV. Via repeat echo, only one child, in the MC group, had two abnormalities noted. The percent of children with persistent defects was not significantly different between the two cohorts (16.7% vs 33.3%, p = 0.64). In total, 4/12 (33.3%) of children with MC had major findings, as opposed to 1/12 (8.3%) in the NC group (Fig 2). The distribution of major defects was statistically different between the NC and MC groups (p = 0.01). There were no severe findings on repeat imaging. The only minor defect, found in the NC group, was mild TR. The remaining defects were persistent PDAs (50.0%), VSDs (33.3%), and an ASD (16.7%). Most major defects in MC children were persistent PDAs (60%). For NC children, the one major defect was a VSD.

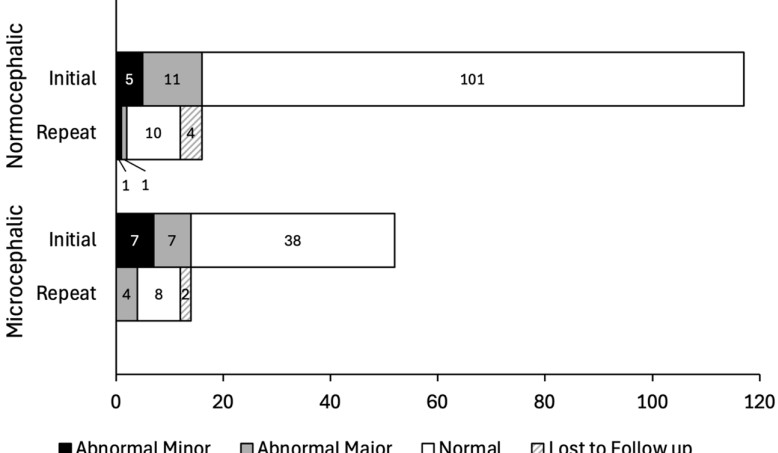

**Fig 2. Echocardiogram Findings by Cephalic Status.** Bar chart depicting breakdown of absolute numbers of initial and repeat echocardiogram findings in normocephalic and microcephalic children. Note that for both cohorts, the majority of defects detected on initial echocardiogram resolved on follow up imaging.

The multivariate logistic regression analysis demonstrates that microcephaly is associated with cardiac anomalies (OR = 3.40, 95% CI 1.15-10.02; p = 0.03). No other clinical/demographic variable had a significant association with echocardiogram findings (Table 3). Further, microcephaly was still associated with higher risk of abnormalities (OR = 6.0, 95% CI 1.03-34.94; p = 0.046) when the analysis was limited to the late group.

## Discussion

Our study outlines the prevalence of structural heart defects in children exposed to ZIKV, with 17.8% of children noted to have cardiac abnormalities in early infancy, mainly in the form of a PDA, VSD, and ASD. In 10.7% of children, the defects were considered major. MC children were noted to have a significantly higher risk of cardiac defects than children with normal head circumference. On repeat assessment, among infants with abnormal initial echocardiograms who returned (24/30), 18 (75.0%) normalized; at the cohort level this equals 18/169 (10.7%) of those imaged and 18/216 (8.3%) of all enrolled. Notably, 6/30 (20.0%) of infants requiring a follow-up echocardiogram did not return, which corresponds to 6/169 (3.6%) of the total analyzed cohort. The proportion with persistent lesions was not significantly different between the MC and NC cohorts, though likely limited by small sample sizes. Importantly, none of the cardiac lesions were

**Table 3. Predictors of Cardiovascular Anomalies in ZIKV-exposed infants.**

| Variable | Unadjusted Odds ratio (+/- 95% CI) | Adjusted Odds ratio (+/- 95% CI) | p-value |
|---|---|---|---|
| Advanced maternal age | 0.56 (0.18-1.73) | 0.83 (0.22-3.08) | 0.77 |
| Cesarean-section | 0.8 (0.35-1.82) | 1.01 (0.38-2.64) | 0.99 |
| Female sex | 1.16 (0.53-2.56) | 1.37 (0.57-3.28) | 0.48 |
| First or second trimester infection | 1.33 (0.45-3.93) | 1.76 (0.55-5.67) | 0.34 |
| Maternal diabetes | 1.16 (0.13-10.8) | 1.87 (0.17-20.99) | 0.61 |
| Maternal hypertension | 0.43 (0.09-1.93) | 0.45 (0.09-2.15) | 0.31 |
| Microcephaly | **2.33 (1.04-5.22)** | **3.40 (1.15-10.02)** | **0.03** |
| Preterm | 1.06 (0.37-3.08) | 1.43 (0.46-4.45) | 0.54 |
| Small for gestational age | 1.49 (0.62-3.59) | 0.58 (0.18-1.88) | 0.36 |

hemodynamically significant, and no child required medical or invasive intervention from a cardiac perspective. Therefore, the clinical implications of these echocardiographic findings remain uncertain, as it is not yet clear whether they translate into meaningful differences in patient outcomes.

This is a follow up study to the analyses by Orofino et al. (2018), which demonstrated a higher incidence of major heart defects in ZIKV exposed infants compared to the general population [24]. Two other studies further corroborate this hypothesis [32, 33]. Specifically, Cavalcanti et al. (2017) observed a higher incidence of structural anomalies (13.5%), especially in the form of septal defects; however, most were not hemodynamically significant [33]. A smaller case series identified CHDs in 2/18 infants, with one having complex disease consisting of pulmonary venous return, total atrioventricular septal defect, and a persistent PDA [32]. For background, CHDs have been reported to occur in approximately 5–7 per 1000 live births in Brazil [28], with a higher prevalence observed in infants with low birth weight [34]. Of note, these figures are derived from passive surveillance systems that preferentially capture more severe cases and under-ascertain milder lesions, particularly in regions with limited diagnostic capacity. Generally, accurate tracking of cardiac defects in Brazil may be difficult given the notable rates of underreporting, especially in the northern and northeastern regions of the country [28]. While the advent of the Unified Health System has increased health care access for Brazilians, there are still challenges in achieving a similar universal healthcare model across the country [35]. We suspect that the regions with the highest rates of underreporting are reflective of the discrepancy in healthcare infrastructure and access to pediatric cardiology services, and thus comparisons to population-based analyses must take this into consideration.

ZIKV is primarily known for its impact on the central nervous system, particularly in neonates, where exposure is associated with CZS, which includes microcephaly and other neurological abnormalities [6]. However, there is emerging evidence suggesting that ZIKV may also affect the heart. Cardiomyopathy has been reported in infant rhesus macaques born to dams infected during pregnancy [36]. An observational study of birth and death records in Brazil detected higher early childhood mortality among children with CZS compared to controls with cardiomyopathy as a leading cause of death [37]. Primary Zika infection has been associated with acute myocarditis leading to a range of symptoms from isolated chest pain to heart failure and arrhythmias [38]. Chronic exposure has been associated with dilated cardiomyopathy and ventricular arrhythmias, possibly secondary to fibrotic changes from long-term inflammation [39,40].

While studies exploring the causal effects of primary ZIKV infection on cardiomyopathies and myocarditis are emerging, the data on the association between antenatal ZIKV exposure and CHDs are limited. Other congenital infections are better understood and may provide guidance. For example, the cardiac defects seen in congenital rubella syndrome (CRS) may be secondary to an inflammatory response from the rubella antigen that leads to vascular injury in cardiac and large vessel fibroblasts [41]. Rubella virus has also been found to affect gene regulation in endothelial cells, which may upregulate inflammatory cytokines and result in cardiac maldevelopment [42]. In fact, the effects of rubella on cardiac development are quite ubiquitous, such that one study found structural heart defects in three-fourths of patients with CRS, mainly in the form of VSDs, ASDs, PDAs, and Tetralogy of Fallot [43]. Another virus under investigation is Coxsackievirus B, as it has been found to alter myocardial proliferative capacity, affect cardiac architecture, and precipitate CHDs [44]. Zika virus has been demonstrated to show cardiotropism via receptors such as ICAM-3 and tyrosine protein kinase 3, which trigger the release of pro-inflammatory markers and cause cellular apoptosis through inflammatory and autoimmune damage [38,45–47]. In mouse studies, ZIKV induces a myocardial immune response via inactivation of the IFNα and β receptor, leading to an increase in intramyocardial proinflammatory cytokines that could serve as the etiology behind myocarditis in humans [48]. In the fetus, exposure to ZIKV and the induction of these inflammatory processes can impact myocardial development and manifest as congenital cardiac and neurologic defects.

Overall, our study found a high prevalence of structural cardiac defects in infants exposed antenatally to ZIKV, and those with more severe clinical sequelae, manifesting as microcephaly, had significantly greater rates during initial screening. We hypothesize that these cardiac and neurologic anomalies may reflect a shared vulnerability secondary to viral or inflammatory insults during organogenesis. The presence of cardiac lesions may suggest a systemic developmental

disturbance; however it is important to acknowledge that the majority resolve and appear to be clinically mild. Regardless, given the increased rates of cardiac findings associated with vertical transmission of ZIKV, we build upon our prior findings, and propose a consideration of the diagnostic criteria of CZS to include CHDs. In hospital settings with robust cardiovascular infrastructure, echocardiogram should be considered in all children with CZS as a screening method to identify structural cardiac lesions. This approach is analogous to screening protocols utilized in other congenital infections, such as long bone radiographic evaluations in syphilis [49] and hearing assessments in cytomegalovirus (CMV) infection [50]. In regions with constrained resources, a more nuanced and context-specific strategy may be warranted. Patients who are normocephalic at birth may follow the same criteria as those established for the general population of mothers and infants, which includes a complete physical exam and measure of oxygen saturation. However, patients with microcephaly should be considered for a post-natal echocardiogram, if resources are permitting. We believe the importance of cardiac screening is underscored by the markedly higher mortality observed in children with CZS compared with those without the syndrome, where the leading causes of death are cardiac in nature [51]. A more comprehensive understanding of the clinical associations of infants who were exposed to ZIKV during pregnancy may help guide screening protocols, early intervention, long-term comprehensive therapeutic care, and public health measures.

This is the first longitudinal study to report follow up echocardiogram findings in infants with confirmed ZIKV exposure. However, there are several limitations to acknowledge. First, we lacked a non-exposed control group, as this was a clinic-based cohort assembled during a public health emergency. Notably, recruiting controls and screening asymptomatic women was not technically feasible during the epidemic given that healthcare resources were heavily strained by the large number of symptomatic referrals. Therefore, our estimates of about 18% abnormal echocardiograms overall, and a higher abnormal rate among MC than NC infants, are benchmarked to population norms rather than infants imaged under identical protocols. Second, selection/referral bias is possible because of 216 exposed infants with postnatal follow-up, 169 (78%) underwent echocardiography, and recruitment at a tertiary pediatric center may result in more complex phenotypes. Third, we acknowledge that follow-up was incomplete, which might have resulted in an underpowered analysis of persistent CV abnormalities: of the 30 infants with initial abnormalities, 24 returned and roughly one quarter had persistent lesions. Additionally, distinguishing physiologic from pathologic findings is challenging given age heterogeneity at imaging (initial 86 ± 80 days; repeat 188 ± 169 days). Finally, we recognize that none of the defects were found to have major clinical repercussions in early childhood, thus the impact of this increased rate of cardiac lesions still needs to be explored.

We designed our follow up study with these limitations in mind. Despite the absence of a non-exposed control group, we included an internal MC-NC comparison under identical protocols, complemented by univariate and multivariable analyses adjusting for gestation, sex, birth weight, delivery mode, trimester of infection, maternal age, hypertension, and diabetes, which still supported an association between MC and cardiac anomalies. Echocardiography was proactively offered to all patients with laboratory-confirmed ZIKV exposures, thus reducing selective imaging by symptom severity. Moreover, imaging acquisition and interpretation were performed by pediatric cardiologists using standardized protocols. Lastly, to minimize the challenges of differentiating between physiologic and pathologic findings, we standardized and prespecified physiologic windows (e.g., PDA in term infants ≤ 15 days and in preterm infants ≤ 3 months; PBS normal in preterm) and performed sensitivity analyses incorporating age at echocardiography. Although these measures do not completely eliminate the limitations described above, they improve the study's internal validity and highlight the increased prevalence of structural cardiac abnormalities in ZIKV-exposed infants, especially among those with MC.

## Conclusions

There was a higher frequency of cardiac defects noted in neonates exposed to ZIKV compared with the general population. Patients with microcephaly were more likely to have abnormal initial echocardiograms compared to children with normocephaly. Therefore, the presence of a congenital cardiac defect could be considered a parameter of CZS given its association with MC. Reassuringly, most of the defects were noted to resolve, and none of the defects were considered

severe. Our findings suggest ZIKV exposure in utero may increase the risk for CHDs secondary to inflammatory mechanisms that still need to be further explored. Therefore, infants with suspected or confirmed ZIKV exposure in utero, especially those with microcephaly, may require close cardiovascular screening. We recognize that some locations in Brazil may lack the ability to perform these studies, thus the local healthcare infrastructure and access to pediatric cardiology services must be considered.

## Supporting information

**S1 Data. Data set for the cohort.**
(XLSX)

## Acknowledgments

We would like to thank the families and children who participated in this study and also the IFF/Fiocruz team for their guidance and support in consolidating and making available the data for this study.

## Author contributions

**Conceptualization:** Brian N. Dang, Karen Kikuta, Trevon Fuller, Patricia Brasil, Dulce H. G. Orofino, Maria Elizabeth L. Moreira, Karin Nielsen-Saines.

**Data curation:** Brian N. Dang, Karen Kikuta, Trevon Fuller, Patricia Brasil, Zilton Vasconcelos, Dulce H. G. Orofino, Maria Elizabeth L. Moreira, Karin Nielsen-Saines.

**Formal analysis:** Brian N. Dang, Karen Kikuta, Trevon Fuller, Patricia Brasil, Zilton Vasconcelos, Dulce H. G. Orofino, Maria Elizabeth L. Moreira, Karin Nielsen-Saines.

**Funding acquisition:** Brian N. Dang, Karin Nielsen-Saines.

**Investigation:** Karen Kikuta, Trevon Fuller, Zilton Vasconcelos, Dulce H. G. Orofino, Maria Elizabeth L. Moreira.

**Methodology:** Brian N. Dang, Karen Kikuta, Trevon Fuller, Patricia Brasil, Zilton Vasconcelos, Dulce H. G. Orofino, Karin Nielsen-Saines.

**Supervision:** Patricia Brasil, Karin Nielsen-Saines.

**Validation:** Brian N. Dang.

**Writing – original draft:** Brian N. Dang, Karen Kikuta.

**Writing – review & editing:** Brian N. Dang, Karen Kikuta, Trevon Fuller, Patricia Brasil, Zilton Vasconcelos, Dulce H. G. Orofino, Maria Elizabeth L. Moreira, Karin Nielsen-Saines.

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
