## [Decision Letter · Decision Letter 0]

10 Oct 2025

Cardiac findings in infants with in-utero exposure to Zika virus – a follow up longitudinal study

Dear Dr. Dang,

Thank you for submitting your manuscript to PLOS Neglected Tropical Diseases. After careful consideration, we feel that it has merit but does not fully meet PLOS Neglected Tropical Diseases's publication criteria as it currently stands. Therefore, we invite you to submit a revised version of the manuscript that addresses the points raised during the review process.

Please submit your revised manuscript within 60 days Dec 09 2025 11:59PM. If you will need more time than this to complete your revisions, please reply to this message or contact the journal office at plosntds@plos.org. Please include the following items when submitting your revised manuscript:

We look forward to receiving your revised manuscript.

Kind regards,

Remi N. Charrel

Academic Editor

Elvina Viennet

Section Editor

Shaden Kamhawi

co-Editor-in-Chief

Paul Brindley

co-Editor-in-Chief

**Additional Editor Comments:**

I advocate the authors to answer specifically all of the comments since the opinion of these reviewers will be seeked for final decision

**Journal Requirements:**

1) Please provide an Author Summary. This should appear in your manuscript between the Abstract (if applicable) and the Introduction, and should be 150-200 words long. The aim should be to make your findings accessible to a wide audience that includes both scientists and non-scientists. Sample summaries can be found on our website under Submission Guidelines:

3) Tables should not be uploaded as individual files. Please remove these files and include the Tables in your manuscript file as editable, cell-based objects. For more information about how to format tables, see our guidelines:

https://journals.plos.org/plosntds/s/tables

4) In the online submission form, you indicated that "The data that support the findings of this study are available from the corresponding author upon reasonable request.". All PLOS journals now require all data underlying the findings described in their manuscript to be freely available to other researchers, either

1. In a public repository

2. Within the manuscript itself

3. Uploaded as supplementary information.

**Reviewers' Comments:**

Reviewer's Responses to Questions

**Key Review Criteria Required for Acceptance?**

**Methods**

-Are the objectives of the study clearly articulated with a clear testable hypothesis stated?

-Is the study design appropriate to address the stated objectives?

-Is the population clearly described and appropriate for the hypothesis being tested?

-Is the sample size sufficient to ensure adequate power to address the hypothesis being tested?

-Were correct statistical analysis used to support conclusions?

-Are there concerns about ethical or regulatory requirements being met?

Reviewer #1: Acceptable

Reviewer #2: The objective of the study is clearly articulated in the final paragraph of the introduction, and the study design is generally appropriate for addressing this aim. However, additional methodological details would strengthen the manuscript and improve clarity.

• The description of the study sample could be expanded. Specifically, in lines 129–132, it should be clarified whether any positive result of the referenced test was sufficient to determine maternal ZIKV exposure.

• At line 137, the authors should explain why some children did not return for the follow-up echocardiogram.

• Lines 137–140: Please specify how many children were excluded due to other congenital infections and genetic syndromes.

• While ethical approval is mentioned, the approval reference numbers from each IRB (INI, IFF/FIOCRUZ, and UCLA) should be provided.

• Were the pediatric cardiologists blinded to ZIKV exposure status?

• Were all prenatal check-ups conducted at the Pediatric Infectious Disease Clinic, or were some patients referred from other centers?

• Were deliveries attended exclusively at the study hospital, or were children born in other hospitals also eligible for referral and inclusion?

Reviewer #3: While the research topic is interesting and relevant, the research question is not clearly defined. It remains ambiguous whether the main goal is to document the frequency of cardiac abnormalities among microcephalic infants, to compare microcephalic vs normocephalic infants, or to investigate or establish a relationship between antenatal ZIKV infection and congenital heart defects (CHD). The authors should clarify the primary and secondary objectives explicitly. The absence of a non-ZIKV-exposed control group seriously limits the capacity to establish whether the prevalence of cardiac findings is truly higher than in the general population. Selection bias is possible since only 169/216 (78%) underwent echocardiography, and no comparison is provided between included and excluded infants. Analytical methods are broadly acceptable (chi-squared, logistic regression), but additional analyses would strengthen the results: (i) provide univariate and multivariate prevalence estimates with crude percentages; (ii) include interaction terms (e.g., microcephaly × trimester of infection); and (iii) present sensitivity analyses stratified by age at echocardiography, as PDA and other lesions may be physiological in early life.

**Results**

-Does the analysis presented match the analysis plan?

-Are the results clearly and completely presented?

-Are the figures (Tables, Images) of sufficient quality for clarity?

Reviewer #1: Need analysis based on time of followup, which is equivalent to infant age.

Reviewer #2: Results

• Clarification is needed regarding the number of symptomatic patients and whether any required medical or surgical treatment.

• Table 3 should include crude ORs so that readers can assess how the estimates changed after adjustment in the multivariate logistic regression.

Reviewer #3: Table 1 should include statistical tests (p-values) to facilitate comparison between microcephalic and normocephalic infants. Table 2 appears structured to explore risk factors for microcephaly, but the more intuitive approach would be to present cardiac abnormalities as the outcome in the columns, with associated factors (microcephaly, SGA, prematurity, trimester of infection, sex) in rows. The table should also indicate denominators and the number of patients affected by each anomaly, not only the number of anomalies. For example: 'VSD: 6 patients (20% of the 30 abnormal patients)'. Figure 1 shows the flow of patients but does not allow direct evaluation of potential selection bias. The persistence of anomalies is based on extremely small numbers (e.g., 4 MC vs 2 NC children), making percentage comparisons misleading. The results should be described with greater caution.

**Conclusions**

-Are the conclusions supported by the data presented?

-Are the limitations of analysis clearly described?

-Do the authors discuss how these data can be helpful to advance our understanding of the topic under study?

-Is public health relevance addressed?

Reviewer #1: Need analysis based on time of followup, which is equivalent to infant age.

Reviewer #2: • The conclusions tend to overstate the results. Clinical significance is not clearly discussed; it is only briefly mentioned that the anomalies did not have major repercussions. This point should be elaborated and integrated into the Results and Discussion sections.

• At lines 273–275, the statement “over 75% of children” is misleading. The data suggest that 3.6% of the original cohort, or 20% of children requiring a follow-up echocardiogram, did not return. The sentence should be revised accordingly.

• It is not appropriate to compare the study’s findings with prevalence estimates derived from a passive surveillance system (line 284). Passive surveillance generally underestimates prevalence and identifies mostly severe cases, while active case ascertainment (as in this study) detects a broader spectrum, including minor anomalies.

• The association between ZIKV exposure and cardiac anomalies appears overstated. Hoffman and Kaplan (2002) showed that while severe congenital heart disease occurs in ~1% of cases, inclusion of minor anomalies raises prevalence to ~7.5%. This context should be acknowledged.

• The recommendation for routine fetal echocardiograms in ZIKV-exposed pregnancies may not be justified. Such practice could add unnecessary stress for families, increase healthcare costs, and may not be clinically warranted, especially given that most anomalies (approximately 75%, as stated) resolved spontaneously. Microcephalic infants warrant closer follow-up, but routine echocardiography for all does not appear supported by the data, particularly since only 7.7% (4/52) of microcephalic patients had persistent anomalies after two echocardiograms, and their clinical significance (symptoms, treatment) is not described.

• The limitations section should be expanded, particularly to address the modest sample size, which may explain why only microcephaly was associated with cardiac anomalies.

Reviewer #3: The conclusions overstate the implications of the findings. The statement that CHD 'should be considered a determinant of CZS' is premature. The evidence is limited to a modest statistical association (OR = 3.40; 95% CI: 1.15–10.02), which is imprecise and based on small numbers. It would be more accurate to frame this as a hypothesis: that severe neurological involvement and cardiac abnormalities may reflect a shared vulnerability of developing organ systems to viral or inflammatory insults during organogenesis. The transient nature of most lesions should also be acknowledged while their initial presence may indicate systemic developmental disturbance, the majority resolve and appear clinically mild.

**Editorial and Data Presentation Modifications?**

Reviewer #1: Line 55 should likely be most of the CV defects resolved over time and at follow up.

Line 62: Suggest: We conducted a follow-up to evaluate cardiologic findings of infants from such mothers.

Line 72: delete (78%) as irrelevant to study

Line 73: Thirty (17.8%) had cardiac anomalies [consisting of ...] that were detected by imaging (?)

Line 77: presence of [persistent?] CV defects

Line 82 and elsewhere: Using the term “most defects” seems misleading since that was not the case in 25% of patients – in a majority of patients might be better

Line 112: “our group” is not a citation – cite first author et al.

Table 1: Instead of having a footnote for abbreviations, incorporate into table itself. Same for Table 3.

Reviewer #2: • Line 68: Define the abbreviation “CV” in the abstract.

• Line 103: RT-PCR abbreviation was already defined previously and need not be repeated.

• Lines 99–101: The description of ZIKV symptoms in adults is not directly relevant to the present study and could be omitted.

• Line 343: Typographical error – “chilhoood” should read “childhood.”

• Figure 2 is misleading when using relative percentages, as it appears that more anomalies were detected during follow-up when in fact the opposite occurred. A revised figure should use absolute numbers and clearly state that 75% of anomalies detected initially resolved on follow-up.

• Line 257: The sentence should be rephrased as: “The multivariate logistic regression analysis suggests that microcephaly is associated with cardiac anomalies” rather than “microcephaly as a predictor of cardiac anomalies.”

Reviewer #3: • Add a clearly defined research question and explicit study objectives in the Introduction.

• Revise Table 1 and Table 2 to include p-values, denominators, and clearer outcome structure.

• Improve figure legends and include 95% confidence intervals in graphical results.

• Correct typographical 'childhoood' l.343).

• Provide a detailed section on study limitations in the Discussion, including: lack of non-exposed controls, selection bias, small sample sizes for persistent lesions, and challenges in interpreting physiologic vs pathologic findings by age and discuss how theses limitations may affect the conclusions if the study

**Summary and General Comments**

Reviewer #1: The study by Dang and colleagues is important and needs to be published, but the manuscript needs considerable revision before being acceptable for PLOS NTD. Below are a number of suggestions that would enhance clarity and readability. The single item that very much needs attention relates to the timing of followup and the conclusion that MC infants have a higher rate of persistence of CV abnormalities than NC infants. The problem is that, unless I somehow missed it, there is no analysis of this conclusion relative to the age of followup. The followup was conducted within 1-12 months (184 ± 139 days), which is a huge interval. Was that interval somewhat similar in MC and NC infants? In other words, the way it is currently written it is unclear whether their might be significant bias in the age at followup between MC and NC infants.

Reviewer #2: This study adds valuable information regarding ZIKV and cardiac anomalies. However, the manuscript requires revision to avoid overstating findings, to provide additional methodological detail, and to strengthen the discussion of clinical relevance and limitations.

Reviewer #3: This manuscript addresses an important and under-researched aspect of congenital Zika syndrome. The longitudinal design and careful definition of exposure are strong features, and the findings are potentially relevant for clinical monitoring and hypothesis generation. However, the study is limited by methodological weaknesses, particularly the absence of a control group, potential selection bias, and small numbers driving the main associations. The interpretation currently overreaches the strength of the evidence. I recommend major revisions to clarify the objectives, strengthen data presentation, and temper conclusions. The addition of a well-documented 'Limitations' section in the Discussion is essential.

PLOS authors have the option to publish the peer review history of their article (what does this mean? ). If published, this will include your full peer review and any attached files.

**Do you want your identity to be public for this peer review?** For information about this choice, including consent withdrawal, please see our Privacy Policy .

Reviewer #1: No

Reviewer #2: No

Reviewer #3: No

**Figure resubmission:**
---

## [Decision Letter · Decision Letter 1]

6 Feb 2026

Dear Dr. Nielsen-Saines,

We are pleased to inform you that your manuscript 'Cardiac findings in infants with in-utero exposure to Zika virus – a follow up longitudinal study' has been provisionally accepted for publication in PLOS Neglected Tropical Diseases.

Best regards,

Remi N. Charrel

Academic Editor

Elvina Viennet

Section Editor

Shaden Kamhawi

co-Editor-in-Chief

Paul Brindley

co-Editor-in-Chief

Reviewer's Responses to Questions

**Key Review Criteria Required for Acceptance?**

**Methods**

-Are the objectives of the study clearly articulated with a clear testable hypothesis stated?

-Is the study design appropriate to address the stated objectives?

-Is the population clearly described and appropriate for the hypothesis being tested?

-Is the sample size sufficient to ensure adequate power to address the hypothesis being tested?

-Were correct statistical analysis used to support conclusions?

-Are there concerns about ethical or regulatory requirements being met?

Reviewer #1: Acceptable

Reviewer #2: The Methods section is clear and sufficiently detailed.

Reviewer #3: The objectives are now appropriate.

**Results**

-Does the analysis presented match the analysis plan?

-Are the results clearly and completely presented?

-Are the figures (Tables, Images) of sufficient quality for clarity?

Reviewer #1: Acceptable

Reviewer #2: The Results are comprehensive and matched the analysis plan. The figures and tables were improved.

Reviewer #3: The resultats are well presented.

**Conclusions**

-Are the conclusions supported by the data presented?

-Are the limitations of analysis clearly described?

-Do the authors discuss how these data can be helpful to advance our understanding of the topic under study?

-Is public health relevance addressed?

Reviewer #1: Acceptable

Reviewer #2: The Discussion and conclusion are thorough and well-structured, covering the main findings, comparisons to prior studies, potential mechanisms, clinical implications, and limitations. The authors clearly addressed how their findings advance the understanding of the topic and its public health relevance.

Reviewer #3: (No Response)

**Editorial and Data Presentation Modifications?**

Reviewer #1: No modifications necessary except perhaps PNTD formatting.

Reviewer #2: (No Response)

Reviewer #3: (No Response)

**Summary and General Comments**

Reviewer #1: The authors did a very credible job of addressing reviewer comments and modifying their manuscript in response.

Reviewer #2: Dear Authors,

I have reviewed the second version of the manuscript entitled “Cardiac findings in infants with in-utero exposure to Zika virus – a follow-up longitudinal study,” submitted to PLOS Neglected Tropical Diseases. I would like to thank you for addressing the comments and recommendations raised in the first review. In its current form, I recommend minor revisions before the manuscript can be considered for publication.

In the abstract, stating an OR of 3.40 is sufficient.

Lines 141–145 (attached document): Should the prevalence of major structural heart defects combined with minor anomalies not be higher than 7.5% or even 10.8%? This sentence requires revision.

Line 183: Spelling mistake. It should read “Congenital.”

I would like clarification on why 47 children did not receive an echocardiogram. Please state the reasons.

In the Statistical Analysis subsection, include the software and version used to perform the planned analyses.

Line 251: In the abstract, it is stated that 30% had microcephaly, while the discussion states 31%. Both numbers must be consistent.

Table 1: The standard spelling is “Cesarean section.”

Lines 351–354: I suggest presenting the information as follows: (OR = 3.40, 95% CI 1.15–10.02; p = 0.03).

Line 500: Change “exposure in-utero” to “exposure in utero.”

Add a Data Availability Statement

These suggestions can also be found in the attached document.

Reviewer #3: (No Response)

PLOS authors have the option to publish the peer review history of their article (what does this mean? ). If published, this will include your full peer review and any attached files.

**Do you want your identity to be public for this peer review?** For information about this choice, including consent withdrawal, please see our Privacy Policy .

Reviewer #1: No

Reviewer #2: No

Reviewer #3: No

---

## [Editor Report · Acceptance letter]

Dear Dr. Nielsen-Saines,

We are delighted to inform you that your manuscript, "Cardiac findings in infants with in-utero exposure to Zika virus – a follow up longitudinal study," has been formally accepted for publication in PLOS Neglected Tropical Diseases.

Best regards,

Shaden Kamhawi

co-Editor-in-Chief

Paul Brindley

co-Editor-in-Chief
